The modulatory effects of gut microbes and metabolites on blood–brain barrier integrity and brain function in sepsis-associated encephalopathy

Li Zhaoying 1 2
Zhang Fangxiang 1
Sun Meisha 1
Liu Jia 1
Zhao Li 1
Liu Shuchun 1
Li Shanshan 1
Wang Bin wangbinhf9999@163.com 1
1 Department of Anesthesiology, Guizhou Provincial People’s Hospital , Guiyang , Guizhou Province , China
2 Institute of Anesthesiology, Guizhou Medical University , Guiyang , Guizhou Province , China
Abdullah Jafri
Electronic publication date: 2023 Mar 28
Publication date: 2023
Volume: 11
Electronic Location ID: e15122
Received 2023 Jan 17; Accepted 2023 Mar 3
Copyright: ©2023 Li et al.
Copyright year: 2023
Copyright holder: Li et al.
License: This is an open access article distributed under the terms of the Creative Commons Attribution License, which permits unrestricted use, distribution, reproduction and adaptation in any medium and for any purpose provided that it is properly attributed. For attribution, the original author(s), title, publication source (PeerJ) and either DOI or URL of the article must be cited.
License URL: https://creativecommons.org/licenses/by/4.0/

Keywords: Sepsis-associated encephalopathy, Microbiome-gut-brain axis, Short-chain fatty acids, Blood–brain barrier, Neuroinflammation

Funding: National Natural Science Foundation of China 81960213 Guizhou Science and Technology Department QKHJC [2020]1Y315 National Natural Science Foundation of China 82260376 This research was funded by National Natural Science Foundation of China, grant number 81960213; Guizhou Science and Technology Department, grant number QKHJC [2020]1Y315; National Natural Science Foundation of China, grant number 82260376. The funders had no role in study design, data collection and analysis, decision to publish, or preparation of the manuscript.

==============================
Background

Intestinal microbiota homeostasis and the gut-brain axis are key players associated with host health and alterations in metabolic, inflammatory, and neurodegenerative disorders. Sepsis-associated encephalopathy (SAE), which is closely associated with bacterial translocation, is a common secondary organ dysfunction and an urgent, unsolved problem affecting patient quality of life. Our study examined the neuroprotective effects of the gut microbiome and short-chain fatty acid (SCFA) metabolites on SAE.

Methods

Male C57BL/6 mice were administered SCFAs in drinking water, then subjected to cecal ligation and puncture (CLP) surgery to induce SAE. 16S rRNA sequencing was used to investigate gut microbiome changes. The open field test (OFT) and Y-maze were performed to evaluate brain function. The permeability of the blood–brain barrier (BBB) was assessed by Evans blue (EB) staining. Hematoxylin and eosin (HE) staining was used to examine intestinal tissue morphology. The expression levels of tight junction (TJ) proteins and inflammatory cytokines was assessed by western blots and immunohistochemistry. In vitro, bEND.3 cells were incubated with SCFAs and then with lipopolysaccharide (LPS). Immunofluorescence was used to examine the expression of TJ proteins.

Results

The composition of the gut microbiota was altered in SAE mice; this change may be related to SCFA metabolism. SCFA treatment significantly alleviated behavioral dysfunction and neuroinflammation in SAE mice. SCFAs upregulated occludin and ZO-1 expression in the intestine and brain in SAE mice and LPS-treated cerebromicrovascular cells.

Conclusions

These findings suggested that disturbances in the gut microbiota and SCFA metabolites play key roles in SAE. SCFA supplementation could exert neuroprotective effects against SAE by preserving BBB integrity.

Introduction

Sepsis is a life-threatening acute organ dysfunction caused by secondary infection (Singer et al., 2016; Evans et al., 2021). The pathogenesis of sepsis involves not only the initial infection and host response but also heterogeneous features of inflammation, the activation of coagulation, the vascular endothelium and the complement system, immune suppression, and alterations in the microbiome (Wiersinga & Vander Poll, 2022). To date, the treatment of sepsis still consists of supportive methods, such as antibiotics, resuscitation, and support of organ dysfunction (Evans et al., 2021). However, the curative effects of these treatments are limited, especially for severe sepsis patients and more vulnerable populations, such as children with chronic disorders (Sazonov et al., 2021). Thus, safer a more effective treatments are urgently needed.

Severe sepsis is often complicated by multiple organ failure, including damage to the central nervous system, which is known as sepsis-associated encephalopathy (SAE). Clinical studies have shown that the incidence of moderate-to-severe cognitive impairment in patients who survive severe sepsis is as high as 70% (Prescott & Angus, 2018). Furthermore, the occurrence of SAE can accelerate the development of multiple organ dysfunction, resulting in prolonged mechanical ventilation, increased intensive care unit (ICU) and total hospital time, and increased mortality (Gofton & Young, 2012).

The pathophysiology of SAE is multifactorial, combining interconnected processes, and is promoted by countless alterations and dysfunctions resulting from sepsis, such as inflammation, neuroinflammation, oxidative stress, reduced brain metabolism, and damage to the integrity of the blood–brain barrier (BBB) (Catarina et al., 2021). The treatment is limited when the cause of SAE is not completely understood. Therefore, studies are critical for a better understanding of its pathophysiology and for the development of new therapies for the prevention and treatment of SAE (Esen et al., 2018).

The BBB, which comprises astrocytes, vascular endothelial cells, pericytes, extracellular matrix and tight junctions (TJs) (zonula occludens-1 (ZO-1) and occludin), controls the homeostasis of water, molecules, and ions in the peripheral circulation and the brain, preventing pathogen, toxin, and immune cell invasion. Sepsis triggers a systemic inflammatory response, releasing endotoxins such as lipopolysaccharide (LPS) and proinflammatory cytokines such as IL-6, IL-1 β, NO, and ROS. This response leads to cellular dysfunction and BBB integrity impairment, and neuronal degeneration and brain edema exacerbate neuroinflammation (Daneman & Prat, 2015; Michinaga & Koyama, 2015). Impaired blood–brain barrier (BBB) function and neuroinflammatory responses are critical features involved in the development of SAE (Widmann & Heneka, 2014; Assimakopoulos et al., 2018). Clinical studies have shown that cytotoxicity or angioedema are the most common features detected by MRI in patients with SAE (Stubbs, Yamamoto & Menon, 2013). In addition, studies have demonstrated that the levels of various TJ proteins are decreased when the BBB is damaged (Danielski et al., 2018), and reversing the changes in ZO-1 and occludin expression can restore BBB integrity and reduce neuroinflammation during sepsis (Liu et al., 2014).

The microbiome, a diverse ecosystem of mostly commensals and mutualists that occupies different niches in the human body, is assumed to interact with most organs of the host (Kundu et al., 2017). The gut microbiome is a dynamic symbiotic system in the gastrointestinal tract, of which bacteria account for more than 99%. Recent studies estimate that there are at least 2776 prokaryotic species that have been isolated from human fecal matter. These have been classified into 11 different phyla, with Proteobacteria, Firmicutes, Actinobacteria, and Bacteroidetes comprising over 90% of the microbiome (Bilen et al., 2018; Hugon et al., 2015; Li et al., 2014), while Fusobacteria and Verrucomicrobia phyla are present in low abundance (Eckburg et al., 2005). A taxonomically diverse intestinal microbiota is associated with the integrity of the epithelial barrier and the maintenance of intestinal metabolic and immune homeostasis. Some species, such as Lactobacillus and Bifidobacterium genera, are considered beneficial and associated with increased intestinal barrier function (Laval et al., 2015). By comparison, overgrowth of Clostridium is often a feature of intestinal microbial dysbiosis and increased intestinal permeability (“leaky gut syndrome”) (Nash et al., 2018). Although the reported broad correlations between obvious compositional changes in the microbiota cannot be used to define a causal role for these correlational observations due to the unique microbiota structure across individuals, previous reports highlighted that the gut might play an indispensable role in the progression of sepsis and multiple organ dysfunction, contributing to epithelial cell apoptosis and barrier impairment and promoting bacterial translocation (Yang et al., 2019; Wang et al., 2015; Rainer et al., 2018; Bajaj et al., 2017).

In fact, gut barrier damage is not the only link between gut bacteria and organ function. There is a complex regulatory network that connects the gut microbiota, and the central nervous system called the microbiome-gut-brain axis (GBA). The GBA is critical in the direct interaction between the gut microbiota and enteric neurons (Barajon et al., 2009; Brun et al., 2013), the production of many chemicals that are required for brain function, and the regulation of the hypothalamic–pituitary–adrenal (HPA) axis (Sudo, 2012). Short-chain fatty acids (SCFAs) are essential metabolites of the intestinal flora and are organic fatty acids with fewer than six carbon atoms that are produced through dietary fiber fermentation in the intestinal tract. Acetic acid, propionic acid and butyric acid are the main SCFAs produced by the intestinal microbiota (Cryan et al., 2019). In the peripheral blood of healthy individuals, these metabolites are present at detectable levels, and more than 95% are absorbed within the colon (Salminen et al., 1998; Topping & Clifton, 2001). An in vitro study demonstrated the regulatory effects of circulating SCFAs on the endothelium of the BBB (Hoyles et al., 2018), suggesting a potential benefit for the treatment of neurological and psychological diseases.

In this study, we attempted to identify the composition signature of the intestinal flora in SAE mice and explore the relationship between the pathophysiology of SAE, gut microbiota changes, and the synthesis of essential SCFAs, with the aim of identifying potential dietary supplements for preventing and treating SAE.

Materials & Methods

Animals

Male C57BL/6 J mice (8–12 weeks, 20 ± 2 g) were purchased from Beijing HFK Bioscience Co., Ltd. and were maintained in the breeding cages of the experimental animal room at 20–22 °C with 55–60% humidity and a light/dark cycle; water and food were available ad libitum during the experimental period. All mice were acclimatized to the experimental environment for 7 days. All animal experimental procedures were reviewed and approved by the Ethics Committee of Guizhou Provincial People’s Hospital (Approval NO. EC Review 2022-010).

Drug treatment and experimental design

One hundred and twenty mice were randomly divided into four groups: sham group, SCFA group, CLP group, and CLP + SCFA group (30 in each group). The sham group underwent a sham operation without SCFA treatment; the SCFA group underwent a sham operation with the short-chain fatty acid (SCFA) treatment; and the CLP group underwent cecal ligation and puncture (CLP) surgery without SCFA treatment. The CLP + SCFA group received CLP surgery and SCFA treatment. SCFA-treated animals were provided drinking water containing sodium acetate (67.5 mM, Sigma Aldrich), sodium propionate (25 mM, Sigma Aldrich), and sodium butyrate (40 mM, Sigma Aldrich) (Sampson et al., 2016) for 14 days before CLP surgery and for 14 days between surgery and sacrifice.

SAE mouse model establishment

The SAE model was established by cecal ligation and puncture (CLP) as previously described (Galli et al., 2005) with slight modifications. Briefly, the mouse was anesthetized with 50 mg/kg pentobarbital by an intraperitoneal injection. A one cm incision was performed on the midline of the anterior abdomen, the cecum was exposed, and the distal 1/3 of the cecum was subjected to ligation (moderate CLP) and punctured twice with a 22 G needle in the ligated segment. Then, the cecum was returned to the abdomen, and adhesion of stool to the surgical incision was avoided. Finally, the skin was sutured. Sham group mice were subjected to the same 1-cm incision on the midline of the anterior abdomen and then sutured without CLP surgery. Postoperatively, all mice were subcutaneously injected with 1 ml of saline for volume resuscitation, and antibiotics (30 mg/kg ceftriaxone sodium + 25 mg/kg clindamycin) were administered at 6 h, 12 h, 24 h, 36 h, and 48 h after the surgery. For the first 3 days, the mice were observed for mortality at least 4 times per day and then twice daily for up to 14 days.

Genomic DNA extraction and 16S rRNA sequencing

Four days after surgery, mice were randomly chosen from the CLP group and the sham group for 16S rRNA sequencing analysis. Each mouse was placed in a separate cage, and 2–3 fresh fecal pellets were collected from each mouse, frozen immediately and stored at −80 °C. The microbial community DNA was extracted using the MagPure Stool DNA KF kit B (Magen, Guangzhou, China) following the manufacturer’s instructions. DNA was quantified with a Qubit Fluorometer by using a Qubit dsDNA BR Assay kit (Invitrogen, Waltham, MA, USA), and the quality was evaluated by running an aliquot on a 1% agarose gel. Afterward, 16S rRNA sequencing and analysis were performed as described previously (Magruder et al., 2019).

16S data processing and analysis

The collected data were filtered for further analysis as described previously (Ren et al., 2022). Briefly, the reads were split into tags by FLASH (Fast Length Adjustment of Short reads, v1.2.11). Then, the tags were clustered into operational taxonomic units (OTUs) by Usearch (V7.0.1090) (Edgar, 2013). Alpha diversity indices and bacterial abundance differences were analyzed based on the relative abundances of known OTUs using the Wilcoxon rank-sum test, and then the Benjamini–Hochberg method was used to adjust the P value. Weighted UniFrac distances were used to analyze beta diversity, and then Adonis and analysis of similarity (ANOSIM) tests were used to evaluate group differences. Then, cluster analysis was performed on the samples, and the similarity of the species composition was judged by the distance between samples. Dissimilarities were visualized by nonmetric multidimensional scaling (NMDS). Hierarchical clustering of samples was performed using UPGMA (unweighted pair group method with arithmetic mean) with unweighted UniFrac as a distance measure. The differences in microbial community composition between groups were compared by linear discriminant analysis (LDA) using LDA Effect Size Tools (V2.0) (Segata et al., 2011). Finally, functional prediction was performed by Phylogenetic Investigation of Communities by Reconstruction of Unobserved States (PICRUSt) 2 to identify enriched KEGG pathways (Langille et al., 2013).

Behavioral tests

Open field test

The OFT was performed on days 4, 7, and 14 after CLP surgery in an open field apparatus (40 cm × 40 cm × 40 cm) divided into three zones, of which the center zone accounted for 25% of the total area. Each mouse was placed in a corner of the field at the beginning of the test and was then recorded for 5 min by a camera above the apparatus. After each trial, the chamber was cleaned with 75% alcohol. The number of entries into the center zone and the time spent in the center zone were recorded by a tracking system with SMART 3.0 (Panlab S.L.U., Barcelona, Spain).

Y-maze test

The Y-maze was used to evaluate hippocampus-dependent spatial working memory and was conducted after the OFT as previously described (Zhang et al., 2018). The Y maze device comprised three identical arms with a removable partition at the junction of the three arms and the central area. The experiment consisted of two phases. During the habituation phase, two arms (start and familiar arm) opened, and the third arm remained blocked (novel arm). Mice were placed at the end of the start arm and explored both the start and familiar arm freely for 5 min; then, they were returned to the waiting cage. The test phase began after 15 min in the waiting cage. Mice were placed at the end of the start arm and allowed to explore the entire maze freely for 5 min. The apparatus was cleaned between the two phases. The percentage of time spent in the novel arm, the percentage of distance traveled in the novel arm, and the frequency of entry into the novel arm were recorded by a tracking system with SMART 3.0 (Panlab S.L.U., Barcelona, Spain).

Evans blue dye leakage assay

Evans blue extravasation was used to evaluate BBB permeability. Briefly, the tail vein (n = 5 per group) was injected with 2% Evans blue dye at a dose of 4 ml/kg. Two hours later, the mice were transcardially perfused with saline and sacrificed. Evan’s blue dye concentrations in the supernatant were assessed using a spectrophotometer (625 nm wavelength). The results are depicted as concentration of Evans blue (ng)/brain tissue (g). To quantify the brain water content, the mice (n = 5 per group) were sacrificed, and the brains were dissected. After being weighed (wet weight), the brain tissues were dehydrated at 56 °C. Forty-eight hours later, the samples were reweighted to obtain the dry weight. The percentage of water was calculated as follows: [(wet weight − dry weight)/wet weight] ×100%. There were cases in which single brains were separated in half, and each half was used to calculate Evans blue levels and the water content.

Hematoxylin and eosin (HE) staining

Morphological changes in intestinal tissue were assessed using HE staining. The steps were performed as previously described (Fischer et al., 2008). Briefly, the intestinal tissue was embedded in paraffin, sectioned, and stained with HE. Pathological lesions were observed using optical microscopy.

Immunohistochemistry (IHC)

Paraffin-embedded intestinal tissues and brain slices containing the hippocampus were used for IHC according to standard protocols as previously described (Han et al., 2019). The expression of occludin (1:200; Abcam, UK) and ZO-1 (1:200, Abcam, Cambridge, UK) and its distribution were examined. After being incubated with primary antibodies at 4 °C overnight, the sections were incubated with horseradish peroxidase (HRP)-conjugated goat anti-rabbit IgG secondary antibodies (Gene Tech (Shanghai) Co., Ltd, Shanghai, China). Images were obtained with an Olympus microscope and analyzed with ImagePro Plus 6.0 software (Media Cybernetics, Rockville, MD, United States).

Western blotting

Intestinal and brain tissues were lysed, and hippocampal tissues were separated. Proteins were collected and quantified as previously described (Wutz et al., 2017). Primary antibodies against ZO-1 (1:1000; Abcam, Cambridge, UK), occludin (Abcam, Cambridge, UK, 1:1000), IL-6 (Abcam, Cambridge, UK, 1:1000), IL-1 β (1:1000; Abcam, Cambridge, UK), TNF- α (1:1000; Abcam, Cambridge, UK), and GAPDH (1:10,000; Abcam, Cambridge, UK) were added for overnight incubation at 4 °C; the samples were then incubated with the secondary antibody (HRP-conjugated goat anti-rabbit IgG, 1:10,000; Abcam, Cambridge, UK) at room temperature for 2 h, followed by chemiluminescent substrate development (Thermo Fisher Scientific). The images were automatically exposed using a chemiluminescence imager (Bio-Rad Laboratories, Hercules, CA, USA). The ratio of the grayscale value of the target protein band to the GAPDH grayscale value (as a normalization control) was calculated to reflect protein expression.

Cerebromicrovascular cell culture and treatment

Mouse brain endothelial bEND.3 cells were obtained from ATCC and maintained in DMEM supplemented with penicillin/streptomycin and FBS. Cells were provided with a mixture of propionate (1 µM), butyrate (1 µM), and acetate (65 µM) for 12 h. Then, Escherichia coli O55:B5 LPS (MedChemExpress, 2 µg/ml) was added for 12 h of stimulation.

Immunofluorescence analysis

bEND.3 cells were cultured on a 24-well plate for 12 h before immunostaining according to standard protocols. Serum was used to block nonspecific antigens at room temperature (20–25 °C) for 1 h. The sections were incubated with primary antibodies against ZO-1 (1:100; Abcam, Cambridge, UK) or occludin (1:100; Abcam, Cambridge, UK) at 4 °C overnight before being incubated with Alexa Fluor 488-conjugated phalloidin (1:1000; Abcam, Cambridge, UK). The samples were mounted using Vectashield with DAPI (cat. no. H-1200, Vector labs). Images were captured using an LSM880 confocal laser scanning microscope (Carl Zeiss Ltd., Cambridge, UK) fitted with 405 and 488 nm lasers. Images were captured with ZEN imaging software (Carl Zeiss Ltd).

Statistical analysis

R software (4.2.0; R Core Team, 2022) was used for statistical analysis. Before analysis, data distribution was assessed by the Shapiro–Wilk test. Student’s t test was used for comparisons between the two groups. Multiple comparison analysis was performed by one-way analysis of variance (ANOVA) followed by an LSD post hoc test. The Wilcoxon rank-sum test was performed as a nonparametric analysis in cases where the data were not normally distributed. The survival rate was analyzed by the Kaplan–Meier method and log-rank test with GraphPad Prism 9.0. All the data are presented as the mean ± standard error of the mean (SEM). The statistical analyses, significance levels and n values are described in the figure or figure legends. In the analysis of the 16S rRNA sequencing results, differences were significant at a false discovery rate (FDR) < 0.05; differences in other results were significant at P < 0.05.

Results

Altered gut microbiota composition and function in mice subjected to CLP

The 16S rRNA sequencing results showed that the diversity of the microbiota was significantly lower in the CLP group than in the sham group, as determined by the Chao1 index (P < 0.05), but there was no significant difference in the Shannon index, Simpson index, or beta diversity (P > 0.05) (Fig. S1). However, the gut microbiota composition differed between the CLP group and the sham group; the patterns of change fell into two clusters (Figs. 1A, 1B). The abundance of bacteria such as Rodentibacter, Acinetobacter, Ruminococcus_torques, Negativibacillus, Bacteroides, Escherichia-Shigella, which are harmful and correlated with infection and cognitive dysfunction (Han et al., 2020; Koizumi et al., 2019; Chen et al., 2022; Sadovnikova et al., 2021), were increased in the feces of septic mice (P < 0.05). By comparison, the abundances of some beneficial genera, such as Catenibacillus, Christensenellaceae_R_7, Alistipes, Phascolarctobacterium, Butyricicoccus and Prevotellaceae_UCG_001, which are positively correlated with SCFA production and cognitive function (Verhaar et al., 2021; Strati et al., 2017; Zhang et al., 2021; Qian et al., 2018), were significantly decreased (P < 0.05) (Fig. 1C). However, significance was lost after correction for multiple testing (FDR > 0.05). Additionally, LDA effect size (LEfSe) analysis showed that the key bacteria in the sham group were Prevotellaceae_UCG_001, Phascolarctobacterium, Acidaminococcaceae, Oscillospiraceae, and others related to SCFA production. In CLP mice, Proteobacteria, Enterobacteriales, Gammaproteobacteria, and Escherichia-Shigella were predominant (Fig. 1D).

Figure 1 The composition of gut microbiota altered in CLP mice.

(A) Non-metric multidimensional scaling (NMDS) distribution and microbiota diversity in CLP and Sham group; (B) hierarchical clustering of feces samples by the Unweighted Pair-group Method with Arithmetic Mean (UPGMA) according to their unweighted UniFrac matrix; (C) the relative abundance of gut microbiota between two groups compared at genus level; (D) cladogram of the most differentially abundant taxa in CLP group and Sham group.

We performed functional annotation and differential analyses to further examine the effects of the altered gut microbiota. The Kyoto Encyclopedia of Genes and Genomes (KEGG) functional annotation results showed that “Biosynthesis of other secondary metabolites”, “Membrane transport” and “infectious deceases: Bacterial” were enriched at level 2 (Fig. 2A). “Fatty acid biosynthesis” was enriched at level 3 (Fig. 2B). Functional difference analysis showed that, at level 2, “neurodegenerative diseases” and “infectious deceases: Bacterial” were stimulated, “glycan biosynthesis and metabolism”, “signaling molecules and interaction” and “signal transduction” were suppressed in the CLP group (FDR < 0.05) (Fig. 2C). At level 3, the “Parkinson’s disease” was significantly enriched , and “fatty acid degradation”, “C5-Branched dibasic acid metabolism”, “sulfur metabolism”, “taurine and hypotaurine metabolism”, “Tyrosine metabolism” and “N-Glycan biosynthesis” were significantly differed in the CLP group (FDR < 0.05) (Fig. 2D), which suggests a state of metabolic disorder.

Figure 2 The function of gut microbiota altered in CLP mice.

(A, B) Relative abundance distinct KEGG categories in the CLP and Sham samples at level 2(a) and level 3(b); (C, D) based on the functional classifications of the KEGG database, the functional categories between two groups compared at level 2(c) and level 3(d). C, CLP group; S, Sham group.

SCFAs alleviate CLP-induced cognitive dysfunction

Based on the functional analysis results, we hypothesized that sepsis altered the intestinal microbiota composition and metabolism in mice and reduced the production of SCFAs, which may be related to sepsis-associated cognitive dysfunction. Thus, we treated septic mice with SCFAs 14 days before and 14 days after surgery (Fig. 3A). We found that SCFA treatment reduced mortality, but the difference was not statistically significant (Fig. S3). The OFT results showed that on the 4th, 7th, and 14th days, the number of entries and total duration in the center area was lower in the CLP group than in the sham group (P < 0.05), indicating that the SAE model was successfully established; on the 7th and 14th days (P < 0.05), these changes were reversed by SCFA treatment (Figs. 3B–3D). In the Y-maze test, the frequency of entry into the novel arm, the proportion of time spent in the novel arm, and the proportion of distance traveled in the novel arm decreased in the CLP group on the 4th, 7th, and 14th days (P < 0.05); these changes were reversed by SCFA treatment (P < 0.05) (Figs. 3E–3H). These results suggest that SCFAs may improve cognitive impairment in septic mice. SCFAs alone had no significant effects.

Figure 3 SCFAs improved CLP-Induced cognitive impairment.

(A) Schematic design of animal experiment procedure and behavior test; (B) representative tracking plot from the OFT; (C, D) time spent in the center (Kruskal-Wallis Test, n = 10 mice per group), the number of entries (One-way ANOVA test, n = 10 mice per group) in the center during the OFT; (E) representative tracking plot from the Y-maze test; (F, H) the frequency of entries in the novel arm (Kruskal-Wallis Test, n = 10 mice per group), percentage of time spent in the novel arm (one-way ANOVA test, n = 10 mice per group), percentage of distance traveled in the novel arm (one-way ANOVA test, n = 10 mice per group) during the Y-maze test (* P < 0.05, ** P < 0.01, *** P < 0.001, Data are presented as means ± SEM).

SCFAs enhance intestinal barrier integrity in CLP mice

To evaluate whether SCFAs positively affect the intestinal barrier in septic mice, we performed HE staining of intestinal tissues. In the CLP group, there was prominent necrosis of intestinal glands in the lamina propria, disintegration of glandular epithelial cells, and interstitial hemorrhage with inflammatory cell infiltration. In the CLP+SCFA group, only local necrosis of the mucosa and submucosa of the colon was observed, and a small number of intestinal glands were accompanied by slight expansion, with some inflammatory cell infiltration in the mucosa and submucosa. The other two groups had no apparent lesions (Fig. 4A).

Figure 4 SCFAs ameliorated intestinal barrier disruption in CLP mice.

(A) Pathological observation of intestinal tissue by HE staining. (Upper, 100 ×; Lower, 400 ×); (B) IHC staining of the occludin and ZO-1 in the hippocampus of the mice (Upper, 100 ×; Lower, 400 ×); (C, D) the relative IHC intensity of occluding (C) and ZO-1 (D) protein (one-way ANOVA test); (E, F) Western blot results and Histogram shows quantified statistical results (one-way ANOVA test). (* P < 0.05, ** P < 0.01, *** P < 0.001, Data are presented as means ± SEM).

We measured ZO-1 and occludin expression in intestinal tissue by immunohistochemistry (Figs. 4B–4D) and Western blotting (Figs. 4E–4F). ZO-1 and occludin levels were significantly lower at the protein level in the CLP group (P < 0.05), and these effects were partially reversed by SCFA treatment (P < 0.05). SCFAs alone had no significant effect on these indicators (P > 0.05).

SCFAs attenuate BBB damage and neuroinflammation in CLP mice

To examine the effect of SCFAs on BBB permeability, we measured the brain water content in mice in each group and injected Evans blue (EB) dye into the mice. BBB permeability was increased in CLP mice (Fig. 5A), as were brain water content and EB dye levels (P < 0.05) (Figs. 5B–5C). In the CLP+SCFA group, these indicators were not significantly different from those in the sham group (P < 0.05).

Figure 5 SCFAs reduced BBB permeability reduced by CLP.

(A) The intact blood–brain barrier in mice. (B) Quantification of blood–brain barrier disruption. (C) Brain water content in infected brains. (One-way ANOVA test, * p < 0.05, ** p < 0.01, and *** p < 0.001, Data are presented as means ± SEM).

IHC (Figs. 6A–6D) and Western blotting (Figs. 6E–6G) demonstrated decreased expression of the TJ proteins ZO-1 and occludin in the brain tissue in the CLP group (P < 0.05), which was partially reversed by SCFA supplementation (P < 0.05). In addition, we measured the expression of inflammatory factors in the hippocampus in each group. The expression of IL-1 β, IL-6, and TNF- α was significantly higher in the CLP group than in the sham group (P < 0.05) and was inhibited by SCFA treatment (P < 0.05). SCFA treatment alone had no significant effect on these indicators (P > 0.05) (Figs. 6H–6K).

Figure 6 SCFAs have a protective effect on BBB destruction in CLP mice.

(A, B) IHC staining of occludin and ZO-1in the hippocampus of the mice (Upper, 200 ×; Lower, 400 ×). (C, D) The relative IHC intensity of occludin and ZO-1 protein (n = 5). (E–G) Western blot results of occludin and ZO-1 in the hippocampus of the mice and Histogram shows statistically quantified results. (H–K) Western blot results of IL-1 β, IL-6, and TNF- α in the mice’s hippocampus and Histogram shows statistically quantified results. One-way ANOVA test, * p < 0.05, ** p < 0.01, and *** p < 0.001, data are presented as means ± SEM.

SCFAs enhance the expression of TJ proteins in vitro

To further verify the effect of SCFAs in vitro, we treated cerebromicrovascular cells with SCFAs before LPS exposure. Then, we measured the expression of ZO-1 and occludin by immunofluorescence (Fig. 7). The fluorescence intensity of ZO-1 and occludin was lower in the LPS group than in the sham group and was partially restored in the LPS+SCFA group, suggesting that SCFAs protect BBB integrity and alleviate SAE.

Figure 7 SCFAs have a protective effect on LPS-induced BBB destruction in sepsis.

(A) Confocal microscopic analysis of expression of the tight junction components occludin and zona occludens-1 (ZO-1) in bEND.3 cells following treatment for 24 h with SCFAs, with or without including 2 ug/ml LPS for the last 12 h incubation. The scale bar (50 µm) applies to all images. Images are representative of at least three independent experiments.

Discussion

It is well recognized that the gut microbiota and metabolites regulate host brain functions and behavior via the GBA, potentially affecting many neurological diseases. However, their role in SAE remains unclear. Based on 16S rRNA sequencing and bioinformatic analysis, we showed that the gut microbiota may be involved in the pathophysiology of SAE through the production of SCFAs, which are key microbial mediators of the gut-brain axis. Furthermore, we demonstrated that SCFA supplementation ameliorated behavioral impairment in SAE mice and speculated that SCFAs can not only improve BBB integrity and prevent the CNS from being damaged by peripheral inflammatory cytokines and toxins but also inhibit excessive microglial activation and the production of proinflammatory cytokines. We provided preliminary validation of this speculation in vitro and in vivo.

SAE is the most common cause of encephalopathy in the medical-surgical ICU. The prevalence of delirium in the ICU can be as high as 32.3% (Salluh et al., 2010). Systemic challenges can trigger CNS inflammation and cognitive dysfunction in animal models of sepsis (Kang et al., 2018). In our current study, we established an SAE model via cecal ligation and puncture (CLP) surgery, which has been widely used for decades as the gold standard sepsis model because it establishes hemodynamics, metabolic changes, and inflammatory progression similar to those observed in human sepsis (Rittirsch et al., 2009). After surgery, we performed behavioral tests to evaluate cognitive function. Considering the weak and vulnerable status of mice subjected to CLP surgery, we selectively used less harmful methods and included percentages as criteria. We used the OFT to measure autonomy, curiosity, and tension and the Y-maze test to assess hippocampal-dependent spatial memory. The results suggested that mice developed behavioral impairments after CLP surgery, and SCFA treatment somewhat alleviated cognitive dysfunction.

The gut-brain axis involves bidirectional communication between the brain and gut. In a normal physiological state, beneficial and pathogenic bacteria are balanced; probiotics account for the majority and can compete with harmful bacteria for nutrients (Gareau, Sherman & Walker, 2010). Given that sepsis is closely related to bacterial infections, which are largely gut derived, we examined whether and how the gut microbiome affects brain function. 16S rRNA sequencing is commonly used for bacterial identification (Song et al., 2005), and the results revealed that there were no significant differences in gut microbiota diversity between the CLP group and the sham group. The composition displayed a feature that differed, which, however, did not withstand statistical correction for multiple testing. Conclusions about clear differences would likely require higher mice numbers or stool samples. Nevertheless, LEfSe analysis revealed changes in the relative abundance of specific bacterial genera. In septic mice, the abundances of pathogenic Proteobacteria, Enterobacteriales, Gammaproteobacteria, Escherichia-Shigella, Lachnoclostridium, and Negativibacillus were significantly increased. These results were consistent with a previous systematic analysis of the gut microbiota of septic patients (Liu et al., 2019).

Notably, many of the genera with decreased abundance in the CLP group may be related to SCFA biosynthesis and metabolism: Butyricicoccus, a butyrate-producing clostridial cluster IV genus (Jeraldo et al., 2016), can initiate anti-inflammatory effects by inducing the production of regulatory T-cells (Narushima et al., 2014); the family Prevotellaceae has been reported to be positively related to SCFA levels in humans (Arumugam et al., 2011) and affects spatial learning and memory by modulating neurotransmission and hippocampal synaptic plasticity (D’Amato et al., 2020); the family Christensenellaceae characteristically carries out a saccharolytic reaction, and its end products include volatile fatty acids (Morotomi, Nagai & Watanabe, 2012); the family Oscillospiraceae, which produces butyrate (Gophna, Konikoff & Nielsen, 2017), is negatively associated with inflammatory diseases (Walters, Xu & Knight, 2014); and the family Acidaminococcaceae can produce propionate (Gallier, Van den Abbeele & Prosser, 2020) and exhibits a reduced abundance in children with autism spectrum disorder (ASD) (Ma et al., 2019). Moreover, functional orthologs annotated by the KEGG Orthology database included fatty acid biosynthesis, membrane transport and the immune system. Based on these results, the gut microbiota may participate in SAE pathology through SCFAs.

Since bacterial translocation plays a key role in the pathology of multiple organ dysfunction in sepsis, we evaluated intestinal barrier integrity by morphological observation and measured TJ protein expression. The results showed that the levels of inflammation and necrosis were reduced, and the expression of the TJ proteins ZO-1 and occludin was elevated by SCFA treatment, indicating a protective effect on the intestinal barrier.

Among the various causes of SAE, BBB impairment and neuroinflammation are key determinants. The presence of vasogenic edema and white matter hyperintensity in patients with SAE is frequently observed by MRI (Stubbs, Yamamoto & Menon, 2013; Ehler et al., 2017), which are indicators of BBB breakdown and have been demonstrated in different animal models of sepsis (Varatharaj & Galea, 2017). An in vivo study showed that LPS injection induced 10 kDa dextran translocation across the BBB in mice, which could be due to TJ damage and microglial activation (Haruwaka et al., 2019). Some bloodborne cytokines pass through the BBB via saturable transporters (Banks et al., 1991; Gutierrez, Banks & Kastin, 1993; Banks, Kastin & Gutierrez, 1994). Subsequently, cytokine receptor activation increases cytokine levels in the brain (Johansson et al., 2013). Proinflammatory cytokines play important roles in learning and memory function in the brain (Yirmiya & Goshen, 2011). Patients with delirium after sepsis and other conditions showed higher IL-6 and IL-8 levels than cognitively healthy patients (Wu et al., 2020; De Rooij et al., 2007; Van Munster et al., 2008), indicating that cytokine-induced toxicity in the brain can worsen cognitive function in septic patients. In our current study, BBB integrity in septic mice was significantly impaired, as demonstrated by TJ protein expression levels in brain tissue. Consistent with previous studies, the expression of the inflammatory cytokines IL-1 β, IL-6, and TNF in hippocampal tissue was upregulated, indicating an exacerbated inflammatory state.

Numerous studies have confirmed that SCFAs can directly or indirectly affect the central nervous system. Compared with that in normal mice, the permeability of the BBB in germ-free mice was significantly higher, and acetate treatment could restore it to a certain extent (Braniste et al., 2014). In vitro, cerebrovascular endothelial cell line (hCMEC/D3) exposure to 1 µM propionate for 24 h resulted in the inhibition of nonspecific inflammatory pathways involved in responses to microbial infection and decreased penetration of E. coli- derived LPS (Hoyles et al., 2018); propionate decreased oxidative stress at the BBB through the NFE2L2 signaling pathway and restored expression of the TJ proteins occludin, claudin 5 and ZO-1, which were reduced by LPS. In addition, in a mouse model of autism, high propionate levels in the brain aggravated autism symptoms. These effects were relieved by butyrate supplementation (MacFabe et al., 2007), suggesting that maintaining the balance of gut microbiota metabolite levels may be important. In vitro studies have shown that SCFAs can cross the BBB via monocarboxylate transporters (MCTs) on endothelial cells (Mitchell et al., 2011; Vijay & Morris, 2014). The average concentration of butyric acid in human brain tissue is approximately 17.0 pmol/mg, and the concentration of propionic acid is approximately 18.8 pg/mg brain tissue (Bachmann, Colombo & Berüter, 1979). Recent studies have detected nucleic acids and proteins derived from bacteria, viruses, and fungi in the brain tissue of Alzheimer’s disease (AD) patients after death, suggesting that microbial metabolites such as SCFAs may be produced by microorganisms that reside in or penetrate the brain (Emery et al., 2017; Readhead et al., 2018).

Moreover, SCFAs are endogenous ligands of orphan G protein-coupled receptors (GPCRs), and intracellular SCFAs can influence gene expression via histone deacetylation (HDAC) inhibition (Dalile et al., 2019), which may directly modulate the differentiation, recruitment, and activation of various immune cells (Rodrigues et al., 2016). A study revealed that acetate, butyrate, and propionate could reduce IL-6 and IL-8 levels in LPS- or TNF α-stimulated HUVECs, and the magnitude depended on the incubation time (Li et al., 2018). The protective effect of immunity reduces the systemic inflammatory response by reducing bacterial and bacterial product translocation, indirectly alleviating the neuroinflammatory response in the brain (Corrêa-Oliveira et al., 2016).

Our study showed that SCFAs could reduce hippocampal expression of IL-1 β, IL-6, and TNF α, indicating an inhibitory effect on neuroinflammation in SAE. Additionally, we found that the reduction in ZO-1 and occludin protein expression in intestinal tissue was reversed by SCFA supplementation, which suggests that SCFAs can protect the intestinal barrier to reduce the translocation of bacteria and bacterial products and the systemic inflammatory response. However, further investigation is needed to determine the neuroprotective mechanisms of SCFAs in SAE. Additionally, our study revealed no significant difference between the SCFA-treated and sham groups, suggesting that SCFA supplementation may play a negligible role in the normal physiological state.

Based on our present study, the protective effect of SCFAs on SAE that involves preserving BBB integrity has been demonstrated. However, there are some limitations to this study. Despite the limitations of relatively small sample sizes for 16S rRNA sequencing, the intake of SCFAs for each mouse could not be normalized. Nevertheless, SCFA supplementation in drinking water seems to be more applicable for clinical treatment and consistent with the physiological process in which the majority of SCFAs are originally fermented and absorbed from the intestine. Moreover, this method protects mice from stress and pain, especially when they are in a vulnerable state after CLP surgery. Another limitation is that we only used male animals, which may introduce investigation bias and limit translational application due to the estrous cycle variability of females. Finally, our study only involved certain phenotypic assays. Considering the minimal amount of SCFAs that cross the BBB, the identified effect may be a secondary effect of some indirect mechanisms. Mechanistic research is required to clarify how SCFAs play a neuroprotective role at the cellular and molecular levels.

Conclusions

In summary, we reveal a drastic change in gut microbiome composition in septic mice, which may be closely related to SCFA metabolism. SCFAs supplementation may be able to ameliorate cognitive impairment and neuroinflammation in septic mice and preserve the integrity of the BBB and intestinal barrier. In summary, maintaining gut microbiota homeostasis and SCFA supplementation may serve as a complementary treatment for SAE.

Supplemental Information

Supplemental Information 1 Microbiota diversity in CLP and Control group

(A) The Chao1 indexed of gut microbiota was obviously lower in CLP group than that in Sham group (P = 0.00794). (B–C) There is no significant difference between two groups in Simpson (B) and Shannon index (C). (D) There is no significant difference between two groups in β diversity.

Click here for additional data file.

Supplemental Information 2 The relative abundance of gut microbiota between two groups compared at the Phylum level (A), Class level (B), Order level (C) and Family level (D)

Click here for additional data file.

Supplemental Information 3 Effect of SCFAs on the survival rate in CLP mice

Values are expressed as survival percentage. (Kaplan Meier method and log-rank test)

Click here for additional data file.

Supplemental Information 4 Results analysed without the sample C1

Click here for additional data file.

Supplemental Information 5 Original gels/blots of Western blotting in figures

We marked the predicted band size of target proteins and some visible molecular weight markers on the images included in the figures. Other replicates are available at Figshare.

Click here for additional data file.

Supplemental Information 6 Full 21 point ARRIVE 2.0 Checklist

Click here for additional data file.

Additional Information and Declarations

Competing Interests

Author Contributions

Animal Ethics

Data Availability

The authors declare there are no competing interests.

Zhaoying Li conceived and designed the experiments, performed the experiments, analyzed the data, prepared figures and/or tables, authored or reviewed drafts of the article, and approved the final draft.

Fangxiang Zhang conceived and designed the experiments, analyzed the data, authored or reviewed drafts of the article, and approved the final draft.

Meisha Sun performed the experiments, analyzed the data, prepared figures and/or tables, and approved the final draft.

Jia Liu performed the experiments, prepared figures and/or tables, and approved the final draft.

Li Zhao performed the experiments, prepared figures and/or tables, and approved the final draft.

Shuchun Liu performed the experiments, prepared figures and/or tables, and approved the final draft.

Shanshan Li performed the experiments, prepared figures and/or tables, and approved the final draft.

Bin Wang conceived and designed the experiments, authored or reviewed drafts of the article, and approved the final draft.

The following information was supplied relating to ethical approvals (i.e., approving body and any reference numbers):

Ethics Committee of Guizhou Provincial People’s Hospital [Approval NO.EC Review 2022-101].

The following information was supplied regarding data availability:

The sequences are available at NCBI SRA: PRJNA924022. The raw measurements are available at figshare: Li, Zhaoying (2023): Raw data.zip. figshare. Dataset. https://doi.org/10.6084/m9.figshare.21875391.v1.

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
