# Peer review of "The modulatory effects of gut microbes and metabolites on blood–brain barrier integrity and brain function in sepsis-associated encephalopathy"

_PeerJ, doi:10.7717/peerj.15122_

## Round 0.1 · original submission · Major Revisions

Please heed all suggestions from the 4 peer reviewers to revise your manuscript especially reviewer 1,2,4.

Reviewer 1 ·

Basic reporting

See section 4

Experimental design

See section 4

Validity of the findings

See section 4

Additional comments

INTRODUCTION
- “Sepsis is a life-threatening acute organ dysfunction caused by secondary infection [1, 2]. It is often complicated by multiple organ failure…”. I think the authors should define better sepsis including general immunopathogenic aspects (see: EBioMedicine. 2022 Dec;86:104363. doi: 10.1016/j.ebiom.2022.104363) and highlight the potential translational importance of their basic research, considering that the treatment of severe sepsis is often difficult and more therapeutic resources are needed, also in more vulnerable populations, such children affected with chronic disorders (refer to: Front Pediatr. 2021 Jun 11;9:672260. doi: 10.3389/fped.2021.672260). Then, the authors can start focusing on the SNC-related aspects, which is the core topic of the study.
- lines 52-65: In my opinion, this paragraph should be more narrative and highlight the general mechanisms of SNC impairment during sepsis. Some details could better fit the discussion, in my opinion.
- Line 66 and after: I think the authors should first define the term microbiota/microbiome and provide the main characteristics of gut microbiome (e.g. Nat Rev Gastroenterol Hepatol. 2017 Oct;14(10):573-584. doi: 10.1038/nrgastro.2017.88). Before introducing the concept of microbiome-gut-brain axis, the authors should highlight the evidence of (gut) microbiome involvement in health and disease, but at the same time they should also highlight the fact that a clear microbiome signature is difficult to define, even in those intestinal diseases in which it is more likely a clear contribution of the microbiome (see celiac disease: Front Pediatr. 2021 Apr 22;9:652208. doi: 10.3389/fped.2021.652208)
- line 82: instead of anticipating the general results, the authors should summarize the main objectives of their study.

MATERIALS AND METHODS
- lines 98-99: please, define here the abbreviations used for the study groups
- please, can you define better “sham operation” here?
- All interventions (including sham operation and CLP) should be summarized and explained, since this is a central aspect.
- statistical analysis is not described in detail and is not clear enough.
- behavioral tests should better explained, too.

RESULTS
- Figure 1 is the driver of the first subsection. I think the explanation of each panel of Figure 1 should expanded in the text of this subsection. The authors should clearly explain what they represented and express the results in detail. Overall, this subsection should be expanded and much more detailed, even by reporting some numerical data.
- The previous comments and request should be applied to all other results sections as well.

DISCUSSION
- The authors should start by highlighting the main experimental findings.
- Then, each specific and main finding should be ordinately and individually discussed, since in the current version the discussion sounds quite dispersive and does not allow to focus on the main points.
- At the end, the authors should discuss the limitations of their experimental approach as regards the translational impact; at the same time, they should also highlight this potential, if any.

CONCLUSION
- The authors tried to summarize the main findings and translational impact here. However, I would suggest moderating the strength of their conclusion.

REFERENCES
- to be updated and completed according to the previous comments and recommendations.

FIGURES
- Figure 1 and 2 could be improved in terms of resolution.
- The captions could be accepted, provided that the authors significantly expand the figures descriptions and their information in the results section, as recommended above.

Reviewer 2 ·

Basic reporting

Short-chain fatty acids (SCFAs) have been reported to be a crucial metabolites in gut-brain axis and increasing evidence are reported to link the role of gut microbiota in producing SCFAs and affecting host health and behaviors. Li et al. investigated the potential role of SCFAs in Sepsis-associated encephalopathy (SAE) and showed that supplementation of SCFAs could rescue the pathogenicity of SAE that potentially caused by lack of SCFA produced from alter gut microbiota. The manuscript is well-structured, and the result is well-demonstrated, thus I would support the publication of this work if the authors could address the points below:


Language: although the manuscript is well-structured, the overall language of the manuscript needs to be thoroughly polished as since some sentences are hard to understand and proof-reading by a native-level speaker would improve the text throughout. Below are some examples:

Line-55, use “The BBB” instead of “BBB”
Line-57, use “homeostasis” instead of “balance”
Line-66, use “complicated” or “dynamic” instead of “powerful”
Line-67, 99% of what?
Line-78-79: please revise: “In the peripheral blood of healthy individuals, these metabolites are present at detectable levels, and more than 95% will be absorbed within the colon”
Line-124: please revise “off-machine data”
Please use “down” instead of “dw” in Figure-1 and Figure-2

Experimental design

Methods:

Line-98: “A hundred and fifty mice were randomly divided into four groups: sham group, SCFA group, CLP group, and CLP + SCFA group (30 in each group)”, However, only 120 mice will be used if there are 4 groups with 30 mice in each group.

Line-119 to 122, DNA extraction method was not mentioned in this section

Validity of the findings

Results:

Figure-1B, please also show the bar-plot at different phylogeny level, such family level. Moreover, sample C1 here looks like an outlier with overgrowth of E.coli and Clostridium, thus I would like to see the result of analysis in Figure-1c,d and Figure-2 without this outlier to confirm the result is not biased to this data point. (This result doesn’t need to be added to the final manuscript though)

Figure-1C: please remove P-value and show FDR in -log10 scale. Also, please show relative abundance in log10 scale. (same for Figure-2c,d)

Line-226 to 227, it is actually hard to see the bar for “Immune system” and “Neurodegenerative diseases” in Figure-2a and 2b. Please revise.

Line-229, the conclusion of “Immune system” here is opposite to Line-226 to 227?

Supplementary Figure-2, please describe how P-value was calculated

Congrats on the promising mouse data in Figure3c to 3h, please indicate how many biological replicates/mice are used here.

Additional comments

Data accession: please deposit your raw data of 16S rRNA sequencing to a public data repository (such as NCBI SRA), thus other researchers could potentially apply it for other analysis.

Reviewer 3 ·

Basic reporting

This is an extremely well written manuscript reporting interesting findings from well designed experiment. I really appreciate that the authors took the steps to test their hypothesis that SCFAs help reduce the inflammatory response caused by sepsis in the brain. The authors incorporated neuro-behavioral findings, biochemical findings, and imaging findings to test their hypothesis. The in vitro bEND.3 cell experiment adds a cherry on the top to identify a pathway through which SCFAs impacted pathological/physiological changes caused by sepsis.

Experimental design

Extremely well thought out experimental design

Validity of the findings

Data interpretation and presentation are very clear. The step-wise logic also helps readers to digest this large piece of manuscript.

Additional comments

Great job and very impressive work!

Reviewer 4 ·

Basic reporting

The abstract is concise and has given the required gist of the article. Adequate references used.

Experimental design

This work would contribute to the future development of complementary therapy for treating Sepsis-associated encephalopathy (SAE).

 In sentence 74 - please expand what HPA is?

Validity of the findings

1. Research is substantial and manuscript is well written with coherent discussion.
2. In this manuscript the authors supported their argument with extensive in-vitro and in-vivo studies. Relevant data/results provided to support Author’s arguments.

---

## Round 0.2 · accepted · Accept

Thank you for the revisions made to the manuscript. It will undergo further galleyproof editing and preparations.

Reviewer 1 ·

Basic reporting

I have no additional major comments

Experimental design

I have no additional major comments

Validity of the findings

I have no additional major comments. The authors sufficiently addressed all the previous points.

Additional comments

I have no additional major comments.